# Age at identification, prevalence and general health of children with autism: observational study of a whole country population

Ewelina Rydzewska,[1] Laura Anne Hughes-McCormack,[1] Christopher Gillberg,[1,2] Angela Henderson,[1] Cecilia MacIntyre,[3] Julie Rintoul,[4] Sally-Ann Cooper [1]

JR since deceased.

¹Institute of Health and Wellbeing, University of Glasgow Mental Health and Wellbeing, Glasgow, UK
²Gillbergcentrum/Gillberg Neuropsychiatry Centre, Göteborgs Universitet/University of Gothenburg, Göteborg, Sweden
³National Records of Scotland, Edinburgh, UK
⁴Scottish Government Health and Social Care Analysis, Edinburgh, UK

**Correspondence to**
Professor Sally-Ann Cooper;
Sally-Ann.Cooper@glasgow.ac.uk

## ABSTRACT

**Objectives** Reported childhood prevalence of autism varies considerably between studies and over time, and general health status has been little investigated. We aimed to investigate contemporary prevalence of reported autism by age, and general health status of children/young people with and without autism.

**Design** Secondary analysis of Scotland's Census, 2011 data. Cross-sectional study.

**Setting** General population of Scotland.

**Participants** All children (n=916 331) and young people (n=632 488) in Scotland.

**Main outcome measures** Number (%) of children/young people reported to have autism and their general health status; prevalence of autism; prevalence of poor health (fair, bad and very bad health); odds ratios (95% confidence intervals) of autism predicting poor health, adjusted for age and gender and OR for age and gender in predicting poor health within the population with reported autism.

**Results** Autism was reported for 17 348/916 331 (1.9%) children aged 0–15, and 7715/632 488 (1.2%) young people aged 16–24. The rate increased to age 11 in boys and age 10 in girls, reflecting age at diagnosis. Prevalence was 2.8% at age 10 (4.4% for boys; 1.1% for girls), and 2.9% at age 11 (4.5% for boys; 1.1% for girls). 22.0% of children and 25.5% of young people with autism reported poor health, compared with 2.0% and 4.4% without autism. Autism had OR=11.3 (11.0 to 11.7) in predicting poor health. Autistic females had poorer health than autistic males, OR=1.6 (1.5 to 1.8).

**Conclusion** Accurate information on the proportion of autistic children and their health status is essential plan appropriate prevention and intervention measures and provide resources for those who may put demand on services designed for autistic people.

## INTRODUCTION

Reports on the prevalence of autism inevitably depend on the criteria used. The concept of autism spectrum disorders has now broadened considerably beyond original descriptions,[1 2] and clinicians also now base their diagnosis on fewer symptoms than

### Strengths and limitations of this study

► Large, whole country population study.
► High response rate of 94%, and a systematic enquiry of everyone regarding autism and their general health status.
► Results are generalisable to other child and young people populations in high-income countries.
► Autism and general health status were self/proxy reported by respondents rather than each person having a clinical assessment.
► Six per cent of records were imputed.

a decade ago.[3] Additionally, there is now increased awareness about autism; hence the reported prevalence of autism has increased. Several systematic reviews have attempted to synthesise research studies on prevalence, with overall prevalence varying, dependent on the studies included, for example, their age-ranges, years the studies were conducted in (and hence criteria), data-collection methods, size and representativeness of included studies. Even when restricted to studies published since 2000, studies selected for inclusion in the reviews have shown wide ranges in reported prevalence.[4–7] Recent reviews are summarised in table 1.

The included age range in studies is likely to be critical in these reported rates, related to the age at which children are diagnosed. This, however, seems to be little investigated. A California, USA, study demonstrated that as well as rates of diagnosis of autism increasing, this was particularly so among preschool children,[8] while a large Swedish study found that the number of autism symptoms in children diagnosed with autism had fallen in children diagnosed at age 7–12 years, but not at age 1–6 years.[3] In the National Survey of Children's Health, USA, 259 (24.6%) of children with autism were diagnosed at <3 years of age,

**Table 1** Examples of findings from systematic reviews of recent studies on childhood/youth prevalence of autism

| Review | | No. of studies | Publication dates of studies | Median prevalence/1000 | Range/1000 |
|---|---|---|---|---|---|
| **Autistic disorder** | | | | | |
| French et al., 2013[4] | Autistic disorder | 26 | 2000–2011 | 2.2 | 0.8–9.4 |
| | Asperger syndrome* | 13 | 1998–2011 | 2.1 | 0.5–2.8 |
| Elsabbagh et al., 2012[5] | Northern European | 16 | 2000–2008 | 1.9 | 0.7–3.9 |
| | Western Pacific | 12 | 2000–2011 | 1.2 | 0.3–9.4 |
| | South East Asia/East Mediterranean | 1 | - | - | - |
| | Americas | 7 | 2001–2010 | 2.2 | 1.1–4.0 |
| | *Overall* | | | 1.7 | 0.3–9.4 |
| Tsai, 2014[6] | | 43 | 2001–2013 | 2.8 | 0.3–19.0 |
| **Pervasive developmental disorder** | | | | | |
| French et al., 2013[4] | | 34 | 2000–2011 | 6.2 | 0.6–26.4 |
| Elsabbagh et al., 2012[5] | Northern Europe | 14 | 2000–2011 | 6.2 | 3.0–11.6 |
| | Western Pacific | 4 | 2004–2011 | - | 1.6–19.0 |
| | South East Asia/East Mediterranean | 4 | 2007–2012 | - | 0.1–10.7 |
| | Americas | 13 | 2001–2010 | 6.5 | 1.3–11.0 |
| | *Overall* | | | 6.2 | 0.1–19.0 |
| Tsai, 2014[6] | | 61 | 2000–2014 | 7.0 | 0.2–26.4 |
| Adak and Halder, 2017[7] | | 25 | 2005–2015 | 9.2 | 0.7–26.4 |

*The authors comment on dubious quality of results.

479 (44.5%) at 3–5 years and 383 (30.9%) at >5 years of age.[9] A review has suggested there remains considerable variation in age at diagnosis.[10] Further current data are clearly needed.

One reason why it is important to understand prevalence of autism is that the health profile of children and young people with autism is thought to differ from that of typically developing children and requires interventions and supports.[11–13] Therefore, these combined factors, that is, knowledge of prevalence and health profile of autistic children, are essential for planning and delivery of services.

Subjective general health status is commonly measured in general population studies, and has been demonstrated to be extremely valid, with a strongly predictive linear gradient across health status (from best to poorest) being associated with subsequent number of medical appointments, hospital admissions and mortality.[14–17] It is, therefore, important to measure if there are general health status differences in children and young people with autism compared with other children. However, in terms of general health status of children and young people with autism, there has been very little research. A study in USA reported parent-rated general health for 895 young people with autism aged 13–25 years at baseline, at five time points across 2001–2009, but did not include a general population comparison group.[18] General health was rated as excellent, very good, good or fair/poor. Fair/poor ratings were reported for 6.6% in 2001, 6.4% in 2003, 7.6% in 2005, 6.1% in 2007 and 6.6% in 2009.[18] A large study presenting data from the 2011–2012 National Survey of Children's Health identified 1188/56 746 children with autism under the age of 18, who were found to have significantly lower log odds of health (−1.30, p<0.001) compared with all other children.[19]

To our knowledge, no other studies have investigated reported general health status of children and young people with autism, nor drawn direct comparisons with the general population. This appears to be a major gap in our knowledge.

This study aimed to investigate, on a large scale (the entire population of a country—Scotland) (1) the prevalence of autism, and age of reporting/identifying autism in childhood and (2) the general health status of children and young people with autism compared with those without autism.

## METHODS
### Procedures
Approval was gained from the Scottish Government for secondary analysis of Scotland's Census, 2011 data under the auspices of a collaborative research project with National Records of Scotland.

## Data source

Scotland's Census, 2011, provides information on the number and characteristics of Scotland's population and households on the census day, 27 March 2011. The census is undertaken every 10 years. It includes the whole Scottish population: people living in communal establishments (such as care homes and student halls of residence) as well as people living in private households. Scotland's Census is one of the few country censuses, and indeed it may be unique, in identifying people with autism. One householder on behalf of all occupants in private households, and manager on behalf of all occupants in communal dwellings, was required to complete the Census information. In the great majority of cases, this was, therefore, a parent of the child/young person. The Census form clearly stated that it is a legal requirement to complete the Census, and that not completing it, or supplying false information, can result in a £1000 fine. The Census team conducted a follow-up of non-responders, and provided help to respond when that was needed, hence the high completion rate of 94%.[20] For 2011, the UK Census Offices endorsed CANCEIS (Canadian Census Edit and Imputation System) as the cornerstone of the 2011 Census Editing Strategy. CANCEIS performs robust, cost-effective editing and imputation while incorporating methodological best practice. The Census team used a Census Coverage Survey, including around 40 000 households, to estimate numbers and characteristics of the missing 6%. The Coverage Survey and Census records were deterministically matched using automated and clerical matching to check for duplicates. Individuals estimated to have been missed from the Census were then imputed using a subset of characteristics from real individuals, including information on their health. The edit and imputation methodology was adapted from the rigorous and systematic guidelines of the Office for National Statistics, which is the UK's largest independent producer of official statistics and the recognised national statistical institute in the UK.[21] Two further Scottish Government reports provide information on the estimation and adjustment process used to produce census population estimates for Scotland[22] as well as full details of the methods and other background information.[23]

## Census variables

People with autism were identified from Census question 20, which asked, 'Do you have any of the following conditions which have lasted, or are expected to last, at least 12 months? Tick all that apply'. There was a choice of 10 response options, which included developmental disorder (eg, autistic spectrum disorder or Asperger's syndrome), learning disability (eg, Down's syndrome), learning difficulty (eg, dyslexia) and mental health condition.

During the methodology development for Scotland's Census, 2011, Ipsos MORI Scotland was commissioned to undertake cognitive question testing on question 20 on long-term health conditions and disabilities. This was to test whether the questions were answered accurately and willingly by respondents, and to identify any changes needed to improve data quality and/or the acceptability of the response options for the Scottish population. Cognitive interviewing is a widely used approach to critically evaluate and improve survey questionnaires.[24] It enables researchers to modify survey material to enhance clarity. Retrospective probing was selected as the most appropriate technique. The questions were tested with 102 participants with a mix of gender, age and health conditions and disabilities (including people with more than one of the conditions), to ensure accurate and willing completion.[25] They included people with autism, intellectual disabilities, dyslexia, dyspraxia, speech impairment, mental health conditions (both milder and more serious) and other long-term conditions. This resulted in a redesign of the question on autism, to 'developmental disorder (eg, autism spectrum disorder or Asperger's syndrome)' in order to accurately capture specifically the data on autism. The questions on the other conditions tested (some of which, from a medical perspective, can be considered as developmental disorders) did not require any modification.

Thus, the choice of wording of the question on autism was informed and carefully considered. The term 'developmental disorder' was used and prompted respondents to reply with regard to only autistic spectrum disorder or Asperger syndrome. The question also distinguished autism from learning disability (which in the UK is synonymous with the international term 'intellectual disabilities'), learning difficulties such as dyslexia, and mental health conditions.

The Census team imputed answers for the 14.7% who did not tick any of the boxes in question 20, based on their free text answers for this question and answers to other health questions in the Census, which increased the completion rate to 97.4%. For the remaining 2.6%, the Census team assumed the most plausible explanation was that the person had no long-term condition but did not see the 'No condition' check box at the end of the question, and hence recorded them as such.[26]

Information on general health status was collected through question 19 which had a 5-point response scale: 'How is your health in general?' (1) very good, (2) good, (3) fair, (4) bad, (5) very bad. Similarly, as for question 20, question 19 was tested during the cognitive question testing during the development of the Census. The question was found to not require any modification.

## Data analysis

We calculated the number and percentage of children reported to have autism, by age and gender. We also calculated the number and percentage of children and young people with and without autism reporting very good, good, fair, bad and very bad health, and compared differences using $\chi^2$ tests. Within the whole population of children and young people in Scotland, we then used a logistic regression to calculate odds ratios (with 95%

CI) of autism predicting a derived, dichotomised variable of poor health (fair, bad or very bad health) versus good health (very good or good health), adjusted for age and gender. Age was categorised into groups of 0–15 years (children), or 16–24 years (youth), with the 0–15 year-olds being the reference group. These age groups were selected as in Scotland full legal capacity, with some limitations, is granted to people aged ≥16. Gender was binary; the reference group was male. We then calculated the ORs of age and gender in predicting poor health within the population with autism. All analyses were conducted with SPSS software V.22.

### Patient and public involvement

The question on autism was included in Scotland's Census, 2011, at the behest of third sector organisations for people with autism. People with autism took part in the cognitive question testing during the planning of the Census. This study was undertaken by the Scottish Learning Disabilities Observatory, which has a specific remit for people with autism; its steering group includes partners from third sector organisations. Results from this study will be disseminated for people with autism in easy-read version via the Scottish Learning Disabilities Observatory website and newsletters.

### RESULTS

### Number (%) of children and young people with autism by age and gender

Scotland's Census, 2011, includes records on 916 331 children aged 0–15 years and 632 488 young people aged

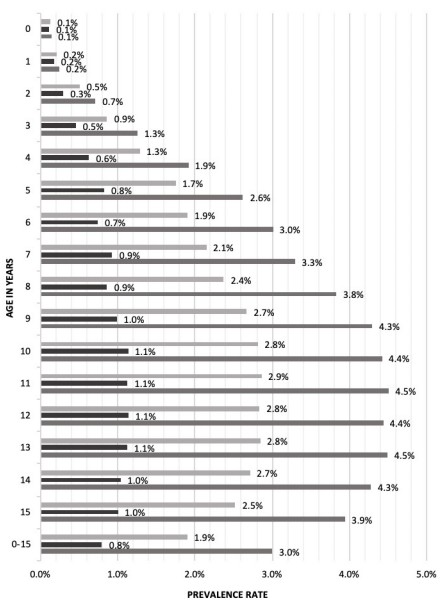

**Figure 1** Identified childhood prevalence of autism by age and gender.

16–24 years. Autism was reported for 17 348 (1.9%) of the children, and 7715 (1.2%) of the young people. Table 2 and figure 1 show the age and gender distribution of the children with and without autism. As expected, there are more males than females with autism; 13 841/17 348 (79.8%) children with autism were male. The rate of reported autism increased to age 11 in boys and age 10

| Table 2 Identified prevalence of childhood autism by age and gender | | | | | | |
|---|---|---|---|---|---|---|
| | **All children** | | | **Children with autism** | | |
| **Age in years** | **Total** | **Female** | **Male** | **Total** | **Female** | **Male** |
| 0 | 58 715 | 28 823 | 29 892 | 76 (0.1%) | 34 (0.1%) | 42 (0.1%) |
| 1 | 59 556 | 29 188 | 30 368 | 126 (0.2%) | 52 (0.2%) | 74 (0.2%) |
| 2 | 58 909 | 28 936 | 29 973 | 301 (0.5%) | 87 (0.3%) | 214 (0.7%) |
| 3 | 58 764 | 28 735 | 30 029 | 509 (0.9%) | 132 (0.5%) | 377 (1.3%) |
| 4 | 56 877 | 27 915 | 28 962 | 730 (1.3%) | 176 (0.6%) | 554 (1.9%) |
| 5 | 55 224 | 26 910 | 28 314 | 966 (1.7%) | 223 (0.8%) | 743 (2.6%) |
| 6 | 55 236 | 26 872 | 28 364 | 1053 (1.9%) | 200 (0.7%) | 853 (3.0%) |
| 7 | 53 786 | 26 172 | 27 614 | 1154 (2.1%) | 244 (0.9%) | 910 (3.3%) |
| 8 | 52 325 | 25 665 | 26 660 | 1243 (2.4%) | 222 (0.9%) | 1021 (3.8%) |
| 9 | 53 046 | 26 022 | 27 024 | 1418 (2.7%) | 257 (1.0%) | 1161 (4.3%) |
| 10 | 55 067 | 26 950 | 28 117 | 1549 (2.8%) | 306 (1.1%) | 1243 (4.4%) |
| 11 | 56 769 | 27 699 | 29 070 | 1623 (2.9%) | 313 (1.1%) | 1310 (4.5%) |
| 12 | 58 656 | 28 412 | 30 244 | 1665 (2.8%) | 324 (1.1%) | 1341 (4.4%) |
| 13 | 59 971 | 29 353 | 30 618 | 1705 (2.8%) | 330 (1.1%) | 1375 (4.5%) |
| 14 | 61 152 | 29 586 | 31 566 | 1658 (2.7%) | 307 (1.0%) | 1351 (4.3%) |
| 15 | 62 278 | 29 987 | 32 291 | 1572 (2.5%) | 300 (1.0%) | 1272 (3.9%) |
| 0–15 | 916 331 | 447 225 | 469 106 | 17 348 (1.9%) | 3507 (0.8%) | 13 841 (3.0%) |

**Table 3** General health status of children and young people with and without autism

| | Age in years | | | | | | | | | | | |
| | 0–15 years n=916 331 | | | | | | 16–24 years n=632 488 | | | | | |
| | Autism | | | Without autism | | | Autism | | | Without autism | | |
| General health | Total 17 348 (100%) | F 3507 (100%) | M 13 841 (100%) | Total 898 983 (100%) | F 443 718 (100%) | M 455 265 (100%) | Total 7715 (100%) | F 1676 (100%) | M 6039 (100%) | Total 624 773 (100%) | F 313 929 (100%) | M 310 844 (100%) |
|---|---|---|---|---|---|---|---|---|---|---|---|---|
| Very good | 7470 (43.1%) | 1291 (36.8%) | 6179 (44.6%) | 758 328 (84.4%) | 376 945 (85.0%) | 381 383 (83.8%) | 3070 (39.8%) | 531 (31.7%) | 2539 (42.0%) | 459 492 (73.5%) | 223 178 (71.1%) | 236 314 (76.0%) |
| Good | 6073 (35.0%) | 1178 (33.6%) | 4895 (35.4%) | 122 814 (13.7%) | 58 499 (13.2%) | 64 315 (14.1%) | 2683 (34.8%) | 605 (36.1%) | 2078 (34.4%) | 137 956 (22.1%) | 75 489 (24.0%) | 62 467 (20.1%) |
| Fair | 2892 (16.7%) | 718 (20.5%) | 2174 (15.7%) | 14 760 (1.6%) | 6800 (1.5%) | 7960 (1.7%) | 1451 (18.8%) | 367 (21.9%) | 1084 (17.9%) | 22 102 (3.5%) | 12 507 (4.0%) | 9595 (3.1%) |
| Bad | 651 (3.8%) | 204 (5.8%) | 447 (3.2%) | 2367 (0.3%) | 1159 (0.3%) | 1208 (0.3%) | 375 (4.9%) | 125 (7.5%) | 250 (4.1%) | 4237 (0.7%) | 2279 (0.7%) | 1958 (0.6%) |
| Very bad | 262 (1.5%) | 116 (3.3%) | 146 (1.1%) | 714 (0.1%) | 315 (0.1%) | 399 (0.1%) | 136 (1.8%) | 48 (2.9%) | 88 (1.5%) | 986 (0.2%) | 476 (0.2%) | 510 (0.2%) |

in girls, being relatively similar across ages 9–15 years for both genders, reflecting the ages at which the autism was diagnosed in the population. Prevalence was 2.8% at age 10 years (4.4% for boys and 1.1% for girls) and 2.9% at age 11 years (4.5% for boys and 1.1% for girls).

## General health

Table 3 shows reported general health status of children and young people with and without autism in Scotland. The children and young people with autism reported poorer health; 22.0% of children and 25.5% of young people with autism reported poor (fair, bad or very bad) general health, compared with only 2.0% of children and 4.4% of young people without autism ($\chi^2$=29 365.6; df=1; p<0.001 for children, and $\chi^2$=7652.1; df=1; p<0.001 for young people). Table 3 shows that the discrepancy between those with and without autism was greater for females than males, for children rather than young people and was even more prominent when comparing bad/very bad health (as opposed to fair/bad/very bad health), for example, 9.1% of girls with autism had bad/

very bad health compared with only 0.4% of girls without autism.

Table 4 shows the results from the regression with the whole population data. Autism had OR=11.3 (95% CI 11.0 to 11.7) in predicting poor health, adjusted for gender and age. Young people were more likely to have poor health than children, as were females. This pattern was also seen within the autistic population, more markedly so for females, and less so for increasing age when compared with the whole population (table 5). Female gender had OR=1.6 (95% CI 1.5 to 1.8), and age 16–24 years had OR=1.2 (95% CI 1.1 to 1.3) in predicting poor health within the autistic population.

## DISCUSSION
### Principle findings and interpretation
We identified the prevalence of reported autism to be 1.9% in children aged 0–15 years overall, and that the reported rate increased with age up to 10 years in girls and 11 years in boys, reflecting the age at which it was identified/diagnosed. Almost all were identified by age 9 years, with the majority before primary school. Prevalence was 2.8% at age 10 years and 2.9% at age 11 years, higher than when the rate is reported for all children overall. This is

**Table 4** OR of autism, age and gender in predicting poor health* in the whole population

| Variable | | OR | 95% CI |
|---|---|---|---|
| Autism | No autism (reference) | – | |
| | Autism | 11.339 | 10.983 to 11.707 |
| Age | 0–15 (reference) | – | |
| | 16–24 | 2.137 | 2.098 to 2.176 |
| Gender | Male (reference) | – | |
| | Female | 1.126 | 1.106 to 1.147 |
| Constant | | 0.020 | |

*Fair, bad or very bad health.

**Table 5** OR of age and gender in predicting poor health* in the population with autism

| Variable | | OR | 95% CI |
|---|---|---|---|
| Age | 0–15 (reference) | – | |
| | 16–24 | 1.206 | 1.133 to 1.284 |
| Gender | Male (reference) | – | |
| | Female | 1.635 | 1.527 to 1.750 |
| Constant | | 0.252 | |

*Fair, bad or very bad health.

of importance when interpreting prevalence studies, as autism in early childhood will clearly be underreported, thus lowering the overall reported childhood prevalence, unless detailed individual assessments are undertaken, which is not realistic in large-scale population-based research. Our study is the only whole-country population study we are aware of to-date to report prevalence of autism using current concepts of the autism spectrum and is highly representative as autism was systematically enquired about for the entire population, with a 94% response rate. Of considerable significance, we report that children/young people with autism were more than 11 times more likely to have poor health than the rest of the population. This inequality was greater for females than males, and more so than in the general population.

### Comparison with existing literature

We found a higher rate of autism than that in the most recent systematic reviews on the subject. This finding most likely reflects that the data are more recent (2011) compared with the most recent reviews, which included data from studies completed a decade earlier, and that we report by year of age, rather than just for all children combined. More comparable studies include the Stockholm Youth Cohort which reported rates of autism in 2011 of 0.40% at age 0–5 years, 1.74% at age 6–12 years, 2.46% at age 13–17 years and 1.76% at age 18–27 years; and of 1.44% at ages 0–17 years overall.[27] The Data Resource Center for Child & Adolescent Health findings for 2014[28] and 2016[9] report higher prevalence of autism at 2.2% (n=243) and 2.5% (n=1131) in all 3–17 year olds but are on a smaller scale. The Autism and Developmental Disabilities Monitoring Network, in 11 sites in the USA, provides estimates of the prevalence of autism in 8-year-old children.[29] In 2014, this varied across sites from 1.3% to 2.9%, with a combined prevalence of 1.7%.[29]

Reported general health was substantially poorer for children and young people with autism compared with the general population. However, there is limited previous research with which to compare our findings; indeed, we believe we are the first to study general health status compared directly with the general population in a whole country population of children and young people with autism. Our findings of poor (fair, bad or very bad) health in 2.0% of children and 4.4% of young people without autism are similar to those reported in a National Health Interview Survey in 2014 which found fair/poor health for 1.6% (n=234) of children aged 0–17 years.[28] However, the study did not report health status separately for children with autism. A further US study reported lower rates of fair/poor health than the 25.5% we found in the young people with autism.[18] It reported fair/poor health in 6.6% in 2001, 6.4% in 2003, 7.6% in 2005, 6.1% in 2007 and 6.6% in 2009 of 895 young people with autism aged 13–25 years at baseline, but did not have a general population comparison group.[18] It also used measures of health not directly comparable with our study, using a 4-point scale of excellent, very good, good and fair/poor

health.[18] Our findings of OR of 11.3 for autism predicting poor general health in the whole population of children and young people are not directly comparable with the findings from the National Survey of Children's Health from 2011 to 2012, due to differences in the scales used, though the results are in the same direction.[19]

Young people with autism had poorer health than children with autism, but the extent of this difference was much less than that seen in the general population. The difference in the extent of influence of age category between the people with and without autism lies in the substantial inequalities in general health that are associated with having autism, regardless of age. Our findings show that children and young people with autism of all ages are more likely to experience poorer general health compared with the rest of the population. We are unable to explain the reasons for this, but note that it is in addition to, and may be related to, their increase in comorbidities compared with other children and young people.[11–13] This requires further investigation.

### Strengths and limitations

This large-scale study covers the whole population of Scotland, and we believe it is currently unique in being a whole country study in which every citizen was systematically enquired about regarding having autism and their general health status. It also had a high completion rate of 94%, suggesting the results are highly representative and likely to be generalisable to other high-income countries. Limitations include the use of the term of 'developmental disorders' in the Census. However, it prompted responses only for the examples of autistic spectrum disorder or Asperger's syndrome, and was tested prior to its use in the Census. Furthermore, the developmental disorders category was distinguished from intellectual disabilities, learning difficulties, and mental health conditions, which are important distinctions. The wording of the question on autism was informed in advance by the cognitive question testing procedure. Therefore, we consider that respondents will have replied accordingly, that is, regarding autism. However, we have no means to check this. Respondents reported whether or not each child/young person was known to have autism rather than each person having an assessment for autism. We are unable to report on the age that each child/young person received their diagnosis; hence we report instead the number of children at each age who have received the diagnosis. They are the proportion at each age who will call on services for children/young people with autism, so this information is important for service planning. Some reporting error is possible, but we are unable to check this. The majority of reports were proxy-reports by parents, but we do not know the extent of proxy versus self-reports for the young people. Neither do we know the extent to which proxy-reporting of general health status compares with an individual's report. The general health status responses were subjective measurements, which might have been influenced by a variety of factors such

as carer burden. It is controversial as to whether autism can be diagnosed in very young children. We found that a small number did report it in the first 2 years. While there may be some reporting error, differences in development in autistic children have been reported to be apparent from as early as 6 months, and widespread by 18 months.[30] The data from this study were collected in 2011, so it will not have captured any changes that have occurred since then. While we described the imputation process, we cannot state with certainty whether the imputed 6% of records contained similar, higher or lower proportion of children and young people with reported autism but note that this missing 6% is a small proportion overall. Imputation of zero by the Census team on the 2.6% of the population with missing data on long-term conditions was not tested, though considered to be the most plausible explanation. Despite these limitations, we believe the results of this study are generalisable to other high-income countries and fill a significant gap in existing research on general health status of children and young people with autism.

## Implications for clinicians

It is essential to have accurate information on the proportion of children and young people who are known to have autism, and their health status, in order to accurately plan appropriate prevention and intervention measures, and provision of resources for those people who may put demand on services designed for people with autism. This requires a full understanding of age differences, and age at diagnosis. The poor general health status observed in the population of children and young people with autism demonstrates a clear need to focus on improvements in healthcare and supports, and the wider determinants of health in this group, which may well differ from the general population.

**Twitter** @ScotLDO

**Acknowledgements** We thank the National Records of Scotland for assisting with the data analysis and dissemination stages of the project. We would also like to pay our last respects to one of our co-authors, Julie Rintoul, who passed away during the peer-review process of this publication. Julie's professionalism, expertise and first and foremost her incredible kindness will leave a long-lasting memory. She will be greatly missed.

**Contributors** ER analysed the data, jointly interpreted it and wrote the first draft of the manuscript. LAH-M, CG and AH jointly interpreted the data, and contributed to the manuscript. CM and JR worked on the Census, jointly interpreted the data and contributed to the manuscript. S-AC conceived the project, interpreted the data and contributed to the manuscript. All authors approved the final version of the manuscript. S-AC is the study guarantor. S-AC confirms the manuscript is an honest, accurate and transparent account of the study being reported, that no important aspects of the study have been omitted and there has been no discrepancies from the study as planned.

**Funding** This study was funded by the Medical Research Council (grant reference MC_PC_17217) and the Scottish Government via the Scottish Learning Disabilities Observatory.

**Competing interests** The funders had no role in the study design, collection, analyses and interpretation of data, in writing the report, nor in the decision to submit the article for publication.

**Patient consent for publication** Not required.

**Ethics approval** Permission to access data was granted by the Scottish Government.

**Provenance and peer review** Not commissioned; externally peer reviewed.

**Data sharing statement** Data are available at http://www.scotlandscensus.gov.uk/ods-web/data-warehouse.html#additionaltab

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
