## [Reviewer comments · BMJ Open]

ARTICLE DETAILS

TITLE (PROVISIONAL)	Age at identification, prevalence, and general health of children with autism - observational study of a whole country population
AUTHORS	Rydzewska, Ewelina; Hughes-McCormack, Laura; Gillberg, Christopher; Henderson, Angela; MacIntyre, Cecilia; Rintoul, Julie; Cooper, Sally-Ann

VERSION 1 - REVIEW

REVIEWER	Bethany Rigles Coleman Institute for Cognitive Disabilities, USA
REVIEW RETURNED	14-Aug-2018

GENERAL COMMENTS	Dear Authors, After reviewing your manuscript entitled, "Age at identification, prevalence, and general health of children with autism – observational study of a whole country population," I have several concerns to report back to you. First, it was unclear why you felt it was important to know the prevalence of autism by age (rather than the age of diagnosis). Age of diagnosis is important to understand in relation to health outcomes, especially because it determines when a child would be eligible to receive services or interventions to reduce the effects of autism. Your study was unable to draw this linkage, however, which made it feel incomplete. Second, your methods section needs substantial work. First, you need to describe the Census information more thoroughly, and emphasize WHO took the survey, if they answered for every member of the household or only one designated member, etc. I would also suggest summarizing important information and referencing the links to more Census information rather than simply providing the links. Additionally, although you describe that the autism question was vetted through cognitive interviewing, as it reads, responding "yes" to having a developmental disability does NOT limit respondents to having autism. There are many types of developmental disabilities. This is a major concern for this paper. It is also of concern that autism cannot be diagnosed prior to age 2, meaning that data for children ages 0-1 is not valid. Further, you need to describe the imputation process you went through (what type of imputation, how many iterations of data,
--

	etc.) in the methods. You mention in the discussion that you did not analyze this data without the imputed data, which is another problem. It is recommended that one always analyze the original data first and then compare it to the imputed data to ensure that nothing out-of-the-ordinary has occurred with the imputed dataset. You also mentioned that the Census team assumed that 2.6% of people did not have a long-term condition (pg. 8, lines 8-11), which is not advisable. We cannot make assumptions about the data—imputation is the best way to recover missing data. You have mentioned several times throughout the paper that no other studies have compared the health status of children/youth with autism to that of the general population. This is not the case (e.g., Rigles, 2017), and I would suggest you return to the literature. I hope these comments are helpful as you move forward with this manuscript.
--	---

REVIEWER	Nicolás Ruiz Robledillo University of Alicante. Department of Health Psychology. Spain.
REVIEW RETURNED	10-Feb-2019

GENERAL COMMENTS	Thank you very much for giving me the opportunity to review this interesting paper. Although I can see the strength of the article (especially the sample size and its representativeness), some issues should be addressed by authors. Introduction More information is needed to justify the inclusion of health status as an aim of the study. Are people with ASD more vulnerable to suffer from health disorders? Data source The data was obtained in 2011, 8 years ago, so it could be a limitation of the study. Procedure One significant limitation is that the obtained prevalence of ASD in the study is originated from a self-reported question, and not from medical records. Other main limitation of the study is the fact that health status has been evaluated by self-report measures. How was measured the health status in individuals with ASD? It is very difficult to evaluate this population through self-reported measures attending the difficulties in written comprehension characteristics of this disorder. If the respondents were caregivers, their perceptions could be subjective and influenced by burden levels. Discussion There is a controversy regarding the ASD diagnosis in children aged 0-2 years. This issue should be addressed by the authors.
--

	Authors should explain or propose specific mechanisms to explain why people with ASD exhibited poorer health in comparison to the general population.
--	---

VERSION 1 – AUTHOR RESPONSE

Reviewers' Comments to Author:

Reviewer: 1

Reviewer Name: Bethany Rigles

Institution and Country: Coleman Institute for Cognitive Disabilities, USA

Please state any competing interests or state 'None declared': None declared

Dear Authors,

After reviewing your manuscript entitled, "Age at identification, prevalence, and general health of children with autism – observational study of a whole country population," I have several concerns to report back to you.

Comment: First, it was unclear why you felt it was important to know the prevalence of autism by age (rather than the age of diagnosis). Age of diagnosis is important to understand in relation to health outcomes, especially because it determines when a child would be eligible to receive services or interventions to reduce the effects of autism. Your study was unable to draw this linkage, however, which made it feel incomplete.

Response: We have added to the discussion of limitation of the paper (Section: Strengths and Limitations, p. 12-13):

'We are unable to report on the age that each child/young person received their diagnosis; hence we report instead the number of children at each age who have received the diagnosis. They are the proportion at each age who will call upon services for children/young persons with autism, so this information is important for service planning.'

Comment: Second, your methods section needs substantial work. First, you need to describe the Census information more thoroughly, and emphasize WHO took the survey, if they answered for every member of the household or only one designated member, etc.

Response: We have amended our description of this so that the information is more prominent in the methods (Section: Data source, page 6):

‘One householder on behalf of all occupants in private households, and manager on behalf of all occupants in communal dwellings, was required to complete the Census information. In the great majority of cases this was, therefore, a parent of the child/young person.’

Comment: I would also suggest summarizing important information and referencing the links to more Census information rather than simply providing the links.

Response: We have summarised the important information and referenced the links as advised. (Sections: Data source, p.6 and Census variables, p. 6-8).

Comment: Additionally, although you describe that the autism question was vetted through cognitive interviewing, as it reads, responding “yes” to having a developmental disability does NOT limit respondents to having autism. There are many types of developmental disabilities. This is a major concern for this paper. It is also of concern that autism cannot be diagnosed prior to age 2, meaning that data for children ages 0-1 is not valid.

Response: From a medical perspective, yes, there are many types of developmental disorders, but the term “Developmental disorder (for example, Autistic Spectrum Disorder or Asperger’s Syndrome)”, was used as understood by the Scottish population to capture autism, and tested by the cognitive question testing procedure prior to its use in the Census. We discuss this in the methods, outlining how the question was refined in the background work to check that its meaning was understood by the Scottish population. We also discuss this issue within the ‘Strengths and limitations’ (p. 12) of our manuscript, and have now made this discussion more prominent in our revision:

‘Limitations include the use of the term of ‘developmental disorders’ in the Census. However, it prompted responses only for the examples of autistic spectrum disorder or Asperger’s syndrome, and was tested prior to its use at the Census. Furthermore, the developmental disorders category was distinguished from intellectual disabilities, learning difficulties, and mental health conditions, which are important distinctions. The wording of the question on autism was informed in advance by the cognitive question testing procedure. Hence, we consider that respondents will have replied accordingly, i.e. regarding autism. However, we have no means to check this.’

We describe the data as recorded in the Census. There are a small number of children in the first two years of life reported to have autism. Whilst this is a controversial point, and there may be some reporting error which we acknowledge in the limitations section (p. 13), we cannot conclude that the data is not valid. We have added and referenced the following point to the strengths and limitations section (p.13):

‘It is controversial as to whether autism can be diagnosed in very young children. We found that a small number did report it in the first two years. Whilst there may be some reporting error, differences in development in autistic children have been reported to be apparent from as early as 6 months, and widespread by 18 months.³⁰’

Comment: Further, you need to describe the imputation process you went through (what type of imputation, how many iterations of data, etc.) in the methods. You mention in the discussion that you did not analyze this data without the imputed data, which is another problem. It is recommended that one always analyze the original data first and then compare it to the imputed data to ensure that nothing out-of-the-ordinary has occurred with the imputed dataset. You also mentioned that the Census team assumed that 2.6% of people did not have a long-term condition (pg. 8, lines 8-11), which is not advisable. We cannot make assumptions about the data—imputation is the best way to recover missing data.

Response: Our study is a secondary analysis of data that was originally processed by the statisticians in the Census team, so we cannot unpick that data in the way suggested. We have, however, improved our description of the imputation process that was used (p. 6): 'For 2011, the UK Census Offices have endorsed CANCEIS (Canadian Census Edit and Imputation System) as the cornerstone of the 2011 Census Editing Strategy. CANCEIS performs robust, cost effective, editing and imputation whilst incorporating methodological best practice. The Census team used a Census Coverage Survey, including around 40,000 households, to estimate numbers and characteristics of the missing 6%. The Coverage Survey and Census records were deterministically matched using automated and clerical matching to check for duplicates. Individuals estimated to have been missed from the Census were then imputed using a subset of characteristics from real individuals, including information on their health. The edit and imputation methodology was adapted from the rigorous and systematic guidelines of the UK's largest independent producer of official statistics and the recognised national statistical institute of the UK.²¹ Two further Scottish Government reports provide information on the estimation and adjustment process used to produce census population estimates for Scotland²² as well as full details of the methods and other background information.²³'

The Census team give a reason for the 2.6% they assume as not having a long-term condition, which we report (p. 7-8). We have also added a reference to the National Records of Scotland further details on this, and refer to it in the limitations section (p.13):

'The Census team imputed answers for the 14.7% who did not tick any of the boxes in question 20, based on their free text answers for this question and answers to other health questions in the Census, which increased the completion rate to 97.4%. For the remaining 2.6%, the Census team assumed the most plausible explanation was that the person had no long-term condition but did not see the 'No condition' check box at the end of the question, and hence recorded them as such.²⁶'

'Imputation of zero by the Census team on the 2.6% with missing data on long-term conditions was not tested, though considered to be the most plausible explanation.'

Comment: You have mentioned several times throughout the paper that no other studies have compared the health status of children/youth with autism to that of the general population. This is not the case (e.g., Rigles, 2017), and I would suggest you return to the literature. I hope these comments are helpful as you move forward with this manuscript.

Response: We have added the reference to this recent, important study by Rigles (2017) on health in children with autism (Section: Introduction, p.5 and Comparison with existing literature, p.12).

Reviewer: 2

Reviewer Name: Nicolás Ruiz Robledillo

Institution and Country: University of Alicante, Department of Health Psychology, Spain.

Please state any competing interests or state 'None declared': None declared

Comment: Thank you very much for giving me the opportunity to review this interesting paper. Although I can see the strength of the article (especially the sample size and its representativeness), some issues should be addressed by authors.

Introduction

Comment: More information is needed to justify the inclusion of health status as an aim of the study. Are people with ASD more vulnerable to suffer from health disorders?

Response: We have added references to support the different health profile of children with autism, and also the reason for investigating general health (Section: Introduction, p. 4).

'One reason why it is important to understand prevalence of autism, is that the health profile of children and young people with autism is thought to differ from that of typically developing children and requires interventions and supports.¹¹⁻¹³ Therefore, these combined factors, i.e. knowledge of prevalence and health profile of autistic children, are essential for planning and delivery of services.

Subjective general health status is commonly measured in general population studies, and has been demonstrated to be extremely valid, with a strongly predictive linear gradient across health status (from best to poorest) being associated with subsequent number of medical appointments, hospital admissions, and mortality.¹⁴⁻¹⁷ It is, therefore, important to measure if there are general health status differences in children and young people with autism compared with other children.'

Data source

Comment: The data was obtained in 2011, 8 years ago, so it could be a limitation of the study.

Response: Yes, we agree, and we have added this to the limitations of the study (p. 13):

'The data from the study was collected in 2011, so it will not have captured any changes that have occurred since then.'

Procedure

Comment: One significant limitation is that the obtained prevalence of ASD in the study is originated from a self-reported question, and not from medical records.

Response: We agree with the reviewer and comment on this limitation of our study in the Section: Strengths and limitations (p. 12):

'Respondents reported whether or not each child/young person was known to have autism rather than each person having an assessment for autism.'

Comment: Other main limitation of the study is the fact that health status has been evaluated by self-report measures. How was measured the health status in individuals with ASD? It is very difficult to evaluate this population through self-reported measures attending the difficulties in written comprehension characteristics of this disorder. If the respondents were caregivers, their perceptions could be subjective and influenced by burden levels.

Response: We acknowledge this limitation of our study in Section: Strengths and Limitations (p. 12), and that the majority of health ratings for the children were provided through proxy reports from their parents (Section: Strengths and limitations, p. 13):

'The majority of reports were proxy-reports by parents, but we do not know the extent of proxy versus self-reports for the young people. Neither do we know the extent to which proxy-reporting of general health status compares with an individual's report. The general health status responses were subjective measurements, which might have been influenced by a variety of factors such as carer burden.'

Discussion

Comment: There is a controversy regarding the ASD diagnosis in children aged 0-2 years. This issue should be addressed by the authors.

Response: Yes this is a controversial point, and there may be some reporting error which we acknowledge in the limitations section by stating the following (p. 13):

'It is controversial as to whether autism can be diagnosed in very young children. We found that a small number did report it in the first two years. Whilst there may be some reporting error, differences in development in autistic children have been reported to be apparent from as early as 6 months, and widespread by 18 months.³⁰'

Comment: Authors should explain or propose specific mechanisms to explain why people with ASD exhibited poorer health in comparison to the general population.

Response: We have added the following to the discussion (Section: Comparison with existing literature, p. 12):

'Our findings show that children and young people with autism of all ages are more likely to experience poorer general health compared to the rest of the population. We are unable to explain the reasons for this, but note that it is in addition to, and may be related to, their increase in comorbidities compared with other children and young people.¹¹⁻¹³ This requires further investigation.'

VERSION 2 – REVIEW

REVIEWER	Nicolás Ruiz Robledillo University of Alicante, Department of Health Psychology, Spain
REVIEW RETURNED	30-Apr-2019

GENERAL COMMENTS	The majority of suggestions of the reviewer have been included as limitations of the study. I would like that authors would have discussed in depth which is the mechanism that could explain the differences in health status between children with ASD and general population, taking into account that is one of the main objectives of the study.
---